# Single cell analysis reveals distinct immune landscapes in transplant and primary sarcomas that determine response or resistance to immunotherapy

Amy J. Wisdom [1], Yvonne M. Mowery [2,3✉], Cierra S. Hong[1], Jonathon E. Himes[1], Barzin Y. Nabet[4,13], Xiaodi Qin[3], Dadong Zhang[3], Lan Chen[5], Hélène Fradin [6], Rutulkumar Patel[2], Alex M. Bassil[2], Eric S. Muise [5], Daniel A. King[4,7], Eric S. Xu [2], David J. Carpenter [2], Collin L. Kent[2], Kimberly S. Smythe [8], Nerissa T. Williams[2], Lixia Luo[2], Yan Ma[2], Ash A. Alizadeh [4,7,9], Kouros Owzar[3,10], Maximilian Diehn [4,9,11], Todd Bradley[12,14] & David G. Kirsch [1,2,3✉]

Immunotherapy fails to cure most cancer patients. Preclinical studies indicate that radiotherapy synergizes with immunotherapy, promoting radiation-induced antitumor immunity. Most preclinical immunotherapy studies utilize transplant tumor models, which overestimate patient responses. Here, we show that transplant sarcomas are cured by PD-1 blockade and radiotherapy, but identical treatment fails in autochthonous sarcomas, which demonstrate immunoediting, decreased neoantigen expression, and tumor-specific immune tolerance. We characterize tumor-infiltrating immune cells from transplant and primary tumors, revealing striking differences in their immune landscapes. Although radiotherapy remodels myeloid cells in both models, only transplant tumors are enriched for activated CD8+ T cells. The immune microenvironment of primary murine sarcomas resembles most human sarcomas, while transplant sarcomas resemble the most inflamed human sarcomas. These results identify distinct microenvironments in murine sarcomas that coevolve with the immune system and suggest that patients with a sarcoma immune phenotype similar to transplant tumors may benefit most from PD-1 blockade and radiotherapy.

[1] Department of Pharmacology and Cancer Biology, Duke University Medical Center, Durham, NC 27708, USA. [2] Department of Radiation Oncology, Duke University Medical Center, Durham, NC 27708, USA. [3] Duke Cancer Institute, Durham, NC 27708, USA. [4] Stanford Cancer Institute, Stanford University, Stanford, CA 94305, USA. [5] Merck & Co., Inc, Kenilworth, NJ 07033, USA. [6] Duke Center for Genomic and Computational Biology, Durham, NC 27708, USA. [7] Division of Oncology, Department of Medicine, Stanford University, Stanford, CA 94305, USA. [8] Fred Hutchinson Cancer Research Center, Seattle, WA 98109, USA. [9] Institute for Stem Cell Biology and Regenerative Medicine, Stanford University, Stanford, CA 94305, USA. [10] Department of Biostatistics and Bioinformatics, Duke University Medical Center, Durham, NC 27710, USA. [11] Department of Radiation Oncology, Stanford University, Stanford, CA 94305, USA. [12] Department of Medicine, Duke University Medical Center, Durham, NC 27710, USA. [13] Present address: Department of Oncology Biomarker Development, Genentech, South San Francisco, CA 94080, USA. [14] Present address: Center for Pediatric Genomic Medicine, Children's Mercy Kansas City, Kansas City, MO 64108, USA. ✉email: yvonne.mowery@duke.edu; david.kirsch@duke.edu

Many cancer patients receive radiation therapy (RT) for palliation or with the intent to cure the irradiated tumor[1]. Preclinical studies using transplanted tumor models demonstrate that focal RT can synergize with immune checkpoint inhibitors to generate systemic antitumor immune responses. In these abscopal responses, RT acts as an in situ vaccine[2–4] to eliminate tumors outside of the radiation field in a T cell- and type I interferon-dependent manner[5–7]. Preclinical studies with transplanted tumors demonstrating high cure rates with checkpoint blockade and RT[8–11] have led to numerous clinical trials[12,13], but emerging results are disappointing[9,14–16].

Here, we administer RT and anti-programmed cell death-1 (PD-1) antibody to mice bearing primary or transplant tumors from a novel high-mutation mouse model of sarcoma[17] to gain insight into mechanisms of response and resistance to immune checkpoint blockade and radiotherapy. Like other studies with transplanted tumors that do not develop in a native microenvironment under immunosurveillance[10,18], we find that transplanted tumors in syngeneic mice are cured by immune checkpoint blockade and RT. However, the identical treatment fails to achieve local control in autochthonous sarcomas from the same model system. We find that autochthonous sarcomas undergo immune editing and repress neoantigenic transcripts. Furthermore, we show that mice that previously developed a primary tumor demonstrate immune tolerance to their own tumors after auto-transplantation, but reject transplanted tumors from other mice. Because auto-transplantation fails to generate a response to anti-PD-1 immunotherapy, our observations support a model in which coevolution of tumors and the immune system generates an immune response that favors tumor tolerance. We find that the immune microenvironment of primary tumors compositionally resembles the majority of human soft tissue sarcomas, which are often immunotherapy-resistant, while transplant tumors model only the most highly inflamed subset of human sarcomas, which are more likely to respond to PD-1 blockade[19]. Using single-cell RNA sequencing and mass cytometric profiling, we profile tumor-infiltrating immune cells from

transplant and primary tumors before and after RT and anti-PD-1 immunotherapy, which reveals marked differences in their immune landscapes. We show that transplant, but not primary, tumors are enriched for activated CD8+ T cells and PD-L1+ macrophages, which are present in human sarcomas that respond to PD-1 blockade[20,21]. These results suggest that patients with a sarcoma immune phenotype similar to transplant tumors may benefit most from PD-1 blockade and radiotherapy.

## Results

**Responses to PD-1 blockade and radiotherapy.** Because human cancers with a higher tumor mutational burden are more likely to respond to immune checkpoint inhibition[22–24], we tested the efficacy of RT and an antibody targeting PD-1 in a high mutational load mouse model of undifferentiated pleomorphic sarcoma (UPS)[17]. We injected the gastrocnemius muscle of $Trp53^{fl/fl}$ mice with an adenovirus expressing Cre recombinase (Adeno-Cre) to delete $Trp53$, followed by injection with the carcinogen 3-methylcholanthrene (MCA). Primary p53/MCA sarcomas developed at the injection site under the selective pressure of the immune system in immunocompetent mice[17]. A cell line from an untreated primary p53/MCA sarcoma was transplanted into the gastrocnemius muscle of syngeneic mice. The resulting tumors were cured by PD-1 blockade and 20 Gy RT (Fig. 1a). However, the same combination treatment failed to cure primary p53/MCA sarcomas (Fig. 1b). Similarly, combined PD-1 and cytotoxic T-lymphocyte-associated protein 4 (CTLA-4) blockade with RT cures transplant p53/MCA sarcomas (Fig. 1c), but this combined immune checkpoint blockade with RT fails to overcome resistance in the primary model (Fig. 1d).

**Tumor-intrinsic immune evasion.** Resistance to immunotherapy can be caused by both an immunosuppressive microenvironment and tumor cell-intrinsic immune evasion mechanisms[25,26]. To identify possible genomic and transcriptomic mechanisms of immune evasion, we performed whole-exome sequencing (WES)

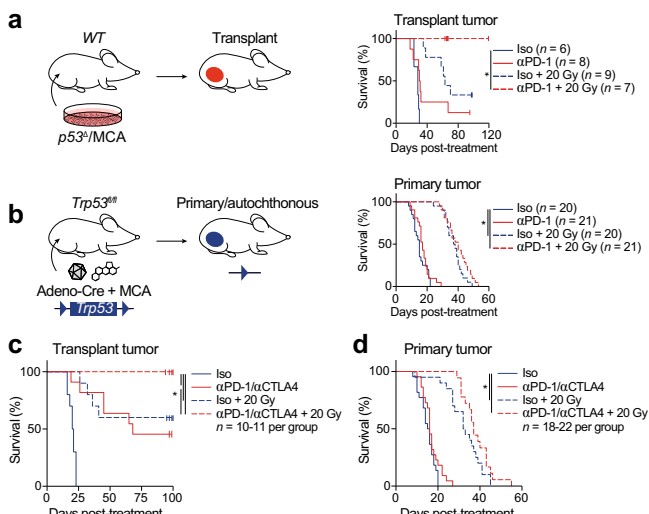

**Fig. 1 Immune checkpoint blockade and radiation therapy cure transplant but not primary sarcomas. a** Transplant tumor initiation by p53/MCA cell injection into the gastrocnemius. Mice were treated with anti (α)-PD-1 (red) or isotype control (blue) antibody and 0 (solid) or 20 (dashed) Gy when tumors reached >70 mm³. **b** Primary sarcoma initiation by intramuscular injection of Adeno-Cre and MCA. Treatment as in panel (**a**). **c** Mice with transplant sarcomas received either both isotype control antibodies with 0 Gy (solid blue, n = 10), αPD-1 and αCTLA-4 with 0 Gy (solid red, n = 11), both isotype control antibodies with 20 Gy (dashed blue, n = 10), or αPD-1 and αCTLA-4 with 20 Gy (dashed red, n = 10). **d** Mice with transplant sarcomas received either both isotype control antibodies with 0 Gy (solid blue, n = 22), αPD-1 and αCTLA-4 with 0 Gy (solid red, n = 22), both isotype control antibodies with 20 Gy (dashed blue, n = 20), or αPD-1 and αCTLA-4 with 20 Gy (dashed red, n = 18). Survival curves estimated using Kaplan–Meier method; pairwise significance determined by log-rank test and Bonferroni correction. *P < 0.05. Source data are provided as a Source data file.

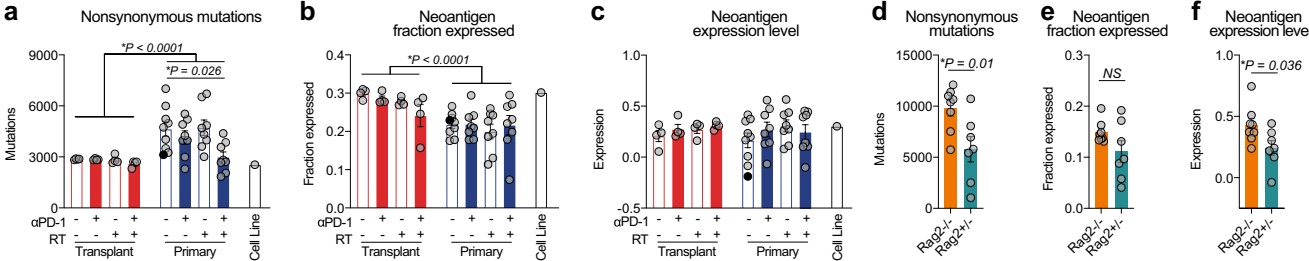

**Fig. 2 Evidence of immune editing at the DNA and RNA levels in primary tumors. a** Nonsynonymous mutations in transplant and primary p53/MCA tumors harvested 3 days after specified treatment. The original primary tumor (filled, black) used to generate the cell line (right) from which the transplant tumors were derived is also shown. $P < 0.001$ for transplant vs primary tumors, $P = 0.026$ for primary tumors with isotype vs primary tumors with anti-PD-1 and 20 Gy. **b** Fraction of neoantigenic mutations expressed by RNA sequencing (>5 mutant and >5 WT reads). $P < 0.001$ for transplant vs primary tumors. **c** Average expression level of genes with neoantigens in transplant tumors, primary tumors, and the cell line quantified as fragments per kilobase of transcript per million mapped reads (FPKM), upper quartile normalized, $\log_2$ transformed. **d** Nonsynonymous mutations in primary p53/MCA tumors from $Rag2^{-/-}$ and $Rag2^{+/-}$ mice harvested when tumor volume reached 70–150 mm³. $P = 0.01$ for $Rag2^{-/-}$ vs $Rag2^{+/-}$. **e** Fraction of neoantigenic mutations expressed by RNA sequencing (>5 mutant and >5 WT reads). **f** Average expression level of genes with neoantigens, calculated as in (**c**). $P = 0.036$ in $Rag2^{-/-}$ vs $Rag2^{+/-}$. For **a–f**, each symbol represents an individual mouse. Mean ± SEM. For **a–c** transplant tumors: $n = 4$; primary tumors: $n = 8$; significance determined by three-way ANOVA with Tukey's multiple comparisons test. For **d–f** $Rag2^{-/-}$ tumors: $n = 8$; $Rag2^{+/-}$ tumors: $n = 7$; significance determined by unpaired two-tailed $t$-test. Source data are provided as a Source data file.

and bulk tumor RNA-seq on primary and transplant p53/MCA tumor samples. In primary and transplant tumors harvested 3 days after treatment with anti-PD-1 or isotype control antibody and 0 or 20 Gy, we compared paired WES data from the tumor and liver of each mouse to identify somatic mutations within each tumor.

Primary tumors harbored more nonsynonymous mutations than transplant tumors (Fig. 2a), indicating that higher tumor mutational burden is not sufficient for sensitivity to immune checkpoint inhibition[22–24]. The parental primary tumor utilized to generate the transplant tumors in Fig. 2 had a similar mutational burden to the transplant tumors, yet only the transplant tumors were cured by anti-PD-1 and RT. In primary tumors, treatment with anti-PD-1 antibody decreased the number of nonsynonymous mutations by ~15%, and the addition of RT resulted in an ~40% decrease in nonsynonymous mutations at 3 days post-treatment (Fig. 2a). To examine whether there was evidence for immune evasion in primary tumors, we computationally examined tumor neoantigens. The fraction of nonsynonymous mutations predicted to be neoantigens was significantly lower in primary tumors than transplant tumors but did not change with treatment (Supplementary Fig. 1a). Despite the decrease in nonsynonymous mutation number in primary tumors after RT and PD-1 blockade (Fig. 2a), the detected fraction of nonsynonymous mutations predicted to be neoantigens was similar across all treatment groups (Supplementary Fig. 1a).

Because checkpoint blockade targets tumor-specific neoantigens[10,27], transcriptional repression of tumor neoantigen expression[26] is an important mechanism for immune escape. Consistent with this observation, primary tumors expressed a smaller fraction of neoantigenic mutations than transplant tumors (Fig. 2b). However, for expressed neoantigens, the corresponding gene expression level did not differ between primary and transplant tumors (Fig. 2c). This suggests that primary tumors may downregulate expression of some neoantigens, while others escape this process. Importantly, the proportion of genes expressed and global gene expression level did not differ between primary and transplant tumors (Supplementary Fig. 1b, c).

These results suggest that in the autochthonous sarcoma model, the immune system imposes selective pressure on the developing tumor. To test this model, we used CRISPR/Cas9 technology[28] to generate primary p53/MCA sarcomas in lymphocyte-deficient $Rag2^{-/-}$ and immune-competent $Rag2^{+/-}$ littermate mice[29]. The

Cas9 protein and the guide RNA targeting *Trp53* were delivered with an adenovirus for transient expression in order to minimize the effect on the immune response to the developing tumor. WES demonstrated that autochthonous p53/MCA sarcomas in $Rag2^{-/-}$ mice harbored nearly twice the number of nonsynonymous mutations compared to primary sarcomas from immune-competent $Rag2^{+/-}$ mice (Fig. 2d). In addition, in primary tumors from $Rag2^{+/-}$ mice, neoantigenic mutations accounted for a smaller proportion of all nonsynonymous mutations (Supplementary Fig. 1d). These findings are evidence for immune editing of the primary tumor by an intact immune system.

We next performed RNA-seq on the same tumors to investigate whether there was evidence for immune-mediated transcriptional downregulation of neoantigens in primary tumors. While the fraction of neoantigens expressed did not differ significantly between sarcomas from $Rag2^{-/-}$ and $Rag2^{+/-}$ mice (Fig. 2e), tumors from immune-competent $Rag2^{+/-}$ mice had significantly lower expression of genes with neoantigenic mutations (Fig. 2f). This transcriptional immune evasion mechanism was specific to neoantigenic mutations, as no differences were seen in global gene expression in tumors from $Rag2^{-/-}$ and $Rag2^{+/-}$ mice (Supplementary Fig. 1e, f). These results further demonstrate the selective pressure of the immune system to promote tumor-intrinsic immune evasion during primary tumor evolution.

**Primary tumors induce immune tolerance.** To test whether the process of in vitro growth and tumor cell transplantation was sufficient to sensitize tumors to RT and immunotherapy, we performed a series of complementary transplantation experiments (Fig. 3). First, we generated primary p53/MCA sarcomas and amputated the tumor-bearing limb when the tumor reached ~70 mm³. We then generated a cell line from each amputated tumor and transplanted this cell line orthotopically into the intact contralateral hind limb of the mouse from which the cell line was derived (i.e., donor mouse), as well as into naive syngeneic mice (Fig. 3a). Tumors grew out with 100% penetrance and significantly decreased latency when transplanted into the donor mice from which the tumor cell lines were derived or T cell-deficient athymic mice, compared to transplantation into immunocompetent naive mice (Fig. 3b, Supplementary Fig. 2). Transplant "self" tumors in donor mice were resistant to tumor cure by anti-PD-1 and RT. When the same tumor cell lines were injected into naive mice and treated with anti-PD-1 and RT, more

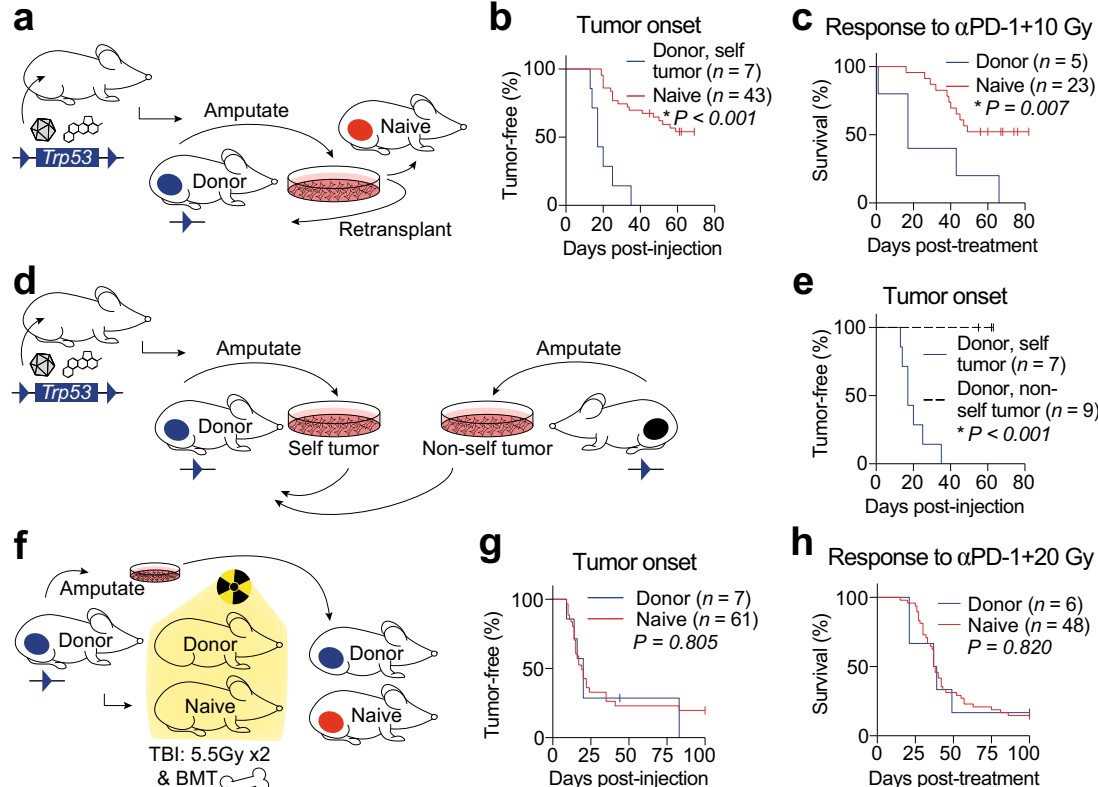

**Fig. 3 Primary tumors induce immune tolerance. a** Transplant tumors generated by injecting cells from a primary tumor into naive mice or donor mice. **b** Time to tumor onset in donor (blue) or naive (red) mice. **c** Survival after αPD-1 and 10 Gy RT. **d** Cell lines from primary tumors in donor ("self") or other ("non-self") mice were injected into donor mice. **e** Time to tumor onset after self (blue) or non-self (black, dashed) cells injected into donor mice. **f** Cell lines from amputated primary sarcomas were injected into donor or naive mice after total body irradiation (TBI) followed by bone marrow transplant (BMT). **g** Time to tumor onset in donor (blue) or naive (red) mice. **h** Survival after αPD-1 and 20 Gy RT in donor (blue) or naive (red) mice that developed tumors. Survival curves estimated using Kaplan–Meier method; significance determined by log-rank test and Bonferroni correction. Source data are provided as a Source data file.

than half of the mice (52%) were cured (Fig. 3c). In contrast to tumor cell lines derived from the same mouse ("self"), "non-self" tumor cell lines were uniformly rejected by naive mice and donor mice, but not by T cell-deficient athymic mice (Fig. 3d, e, Supplementary Fig. 2).

To test whether this immune tolerance could be overcome by an immune system that did not coevolve with the primary tumor, we again generated independent primary p53/MCA sarcomas, amputated the tumor-bearing limb from donor mice when the tumors reached ~70 mm³, and generated cell lines from each amputated tumor. After the donor mice had recovered from surgery, donor and naive mice received a lethal dose of total body irradiation (TBI) and rescue by bone marrow transplant (BMT). After recovery, "self" tumor cells were transplanted into donor mice. Age-matched naive mice also received a simultaneous injection of the same tumor cells (Fig. 3f). Bone marrow transplantation of donor mice restored their ability to reject tumor transplants to the level of age-matched naive mice (Fig. 3f, g, Supplementary Fig. 2). For mice that did develop tumors, donor and naive mice had similar responses to anti-PD-1 and RT (Fig. 3h, Supplementary Fig. 2). Taken together, these data demonstrate that primary p53/MCA tumor development induces immune tolerance to tumor cells that coevolve with their immune system, which cannot be overcome by the immunostimulatory effects of increased neoantigen expression from in vitro growth, tumor cell injection, or treatment with RT and immunotherapy. However, replacing a tolerized immune system by bone marrow transplantation to provide a naive immune system prior to

immunotherapy treatment, an approach that has shown efficacy in patients after the loss of immune-mediated antitumor activity[30], restores immunity to a transplanted tumor to the same level seen in mice with a naive immune system.

**Transcriptional parallels of mouse and human sarcomas.** To identify the major transcriptional differences between primary and transplant sarcomas, we analyzed bulk tumor RNA harvested 3 days after treatment with either 0 or 20 Gy and anti-PD-1 or isotype control antibody. Notably, principal components analysis showed that tumor model type (primary vs. transplant) was the major factor driving transcriptional differences, rather than treatment with radiation and/or anti-PD-1 therapy (Supplementary Fig. 3a). Comparing the gene expression differences in transplant and primary sarcomas revealed that, within the many differentially expressed genes (Supplementary Fig. 3b, c), transplant tumors exhibited enrichment of immune-related pathways, even without treatment (Isotype + 0 Gy) (Fig. 4a). These findings suggested that a more highly inflamed immune microenvironment was present within transplant tumors compared to primary tumors.

To further explore the specific differences in the immune microenvironment of primary and transplant sarcomas, we performed CIBERSORTx, a method to enumerate cellular fractions of a mixed population by deconvolution of gene expression data from a mixed cell population[31]. We compared gene expression data from both primary and transplant tumors to

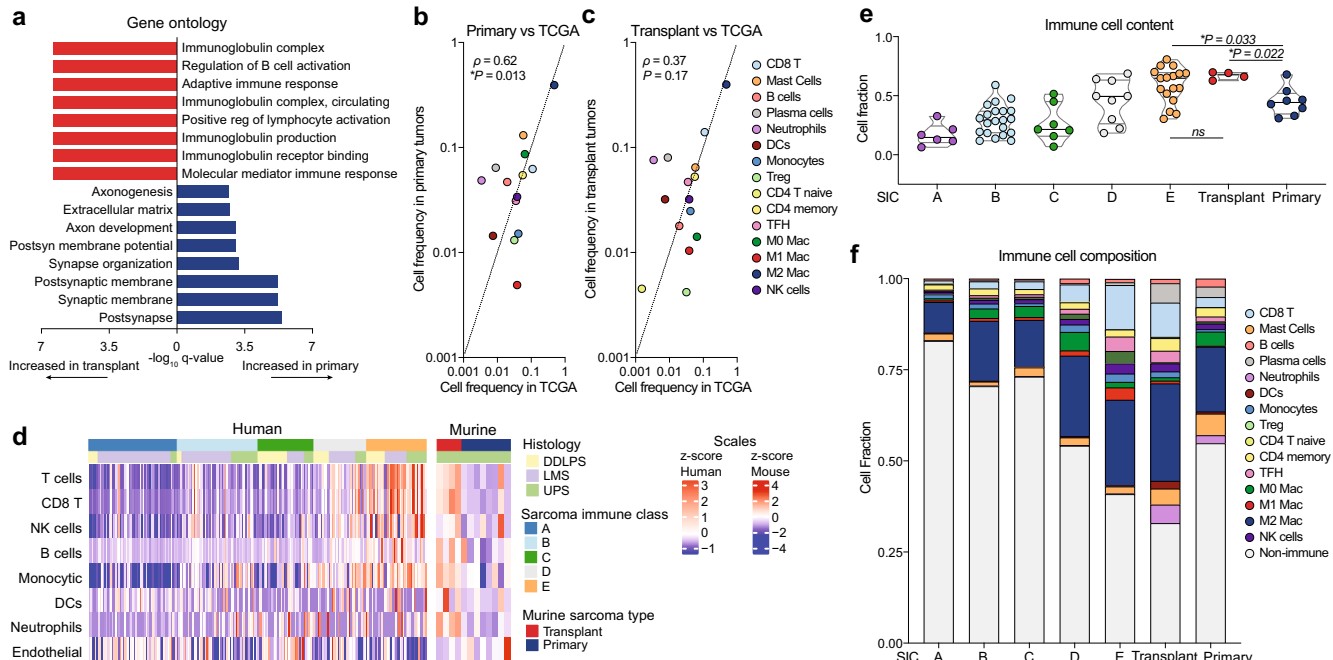

**Fig. 4 Transcriptional signatures in mouse and human sarcomas. a** Gene ontology analysis of differentially regulated processes in primary (blue) versus transplant (red) sarcomas after isotype treatment. **b** Spearman correlation analysis of bulk tumor RNA from primary tumors ($n = 8$) and untreated human UPS (TCGA, $n = 87$). **c** Spearman correlation of bulk tumor RNA from transplant tumors ($n = 4$) and untreated human UPS (TCGA, $n = 87$). **d** Cellular composition of TCGA sarcomas (DDLPS, dedifferentiated liposarcoma; LMS, leiomyosarcoma; UPS, undifferentiated pleomorphic sarcoma) ($n = 218$) separated by previously defined sarcoma immune class (SIC, left) and murine primary sarcomas (right). Sample sizes: SIC A ($n = 57$), SIC B ($n = 52$), SIC C ($n = 36$), SIC D ($n = 34$), SIC E ($n = 39$), primary ($n = 8$), and transplant ($n = 4$). **e** Immune cell content of TCGA sarcomas by sarcoma immune class (SIC) and murine sarcomas. Sample sizes: SIC A ($n = 6$), SIC B ($n = 21$), SIC C ($n = 7$), SIC D ($n = 9$), SIC E ($n = 17$), primary ($n = 8$), and transplant ($n = 4$). Significance determined by two-sided Wilcoxon test. **f** Immune cell composition of TCGA sarcomas by sarcoma immune class (SIC) and murine sarcomas. In **b**–**f**, cell proportions enumerated by CIBERSORTx. Source data are provided as a Source data file.

human UPS from The Cancer Genome Atlas (TCGA, Supplementary Data 1)[32]. The immune composition from primary, but not transplant, tumors was significantly correlated with signatures of human UPS (Fig. 4b, c).

Petitprez et al. recently described an immune-based classification of soft tissue sarcoma patient samples defined by the composition of the tumor microenvironment, identifying five distinct phenotypes: immune-low (Classes A and B), immune-high (Classes D and E), and highly vascularized (Class C)[19]. The most highly inflamed sarcoma immune class (SIC) E expressed high levels of MHC-I, was enriched for a cytotoxic lymphocyte signature, and was associated with a high response rate to PD-1 blockade and improved overall survival[19]. Using CIBERSORTx, we analyzed TCGA gene expression data from human UPS samples and confirmed the presence of high levels of immune infiltrate in human sarcomas previously classified as SIC E tumors by Petitprez et al. Untreated transplant tumors from mice resembled the highly inflamed SIC E sarcomas, while primary tumors resembled the less inflamed sarcoma immune classes (Fig. 4d, Supplementary Fig. 3d). Indeed, the overall immune cell content and the fraction of specific immune cell types of transplant tumors was most similar to the SIC E tumors, which are most sensitive to anti-PD-1 therapy, while primary tumors were more similar to the other human sarcoma immune classes (Fig. 4e, f).

**Primary and transplant tumors create unique immune microenvironments.** Tumor-infiltrating myeloid cells can promote tumor progression through surface expression of immune checkpoint molecules such as PD-L1 and by production of anti-inflammatory

cytokines that induce immune suppression and resistance to checkpoint inhibition[33–35]. We performed CIBERSORTx to profile the myeloid cell compartments of untreated primary and transplant p53/MCA tumors and compared them to the various sarcoma immune classes in patient tumors. Transplant tumors, like SIC E tumors, were enriched for M2 macrophages, while primary tumors, like the immune-low SIC tumors, had significantly lower expression of M2 macrophage genes (Fig. 5a).

Interestingly, sarcoma patients with an objective response to pembrolizumab had a significantly higher percentage of tumor-associated macrophages expressing PD-L1 at baseline compared with non-responders[20]. To test whether PD-L1+ macrophages differed between primary and transplant tumors, we used a panel of 37 heavy metal-conjugated antibodies to analyze independent tumor samples by mass cytometry (CyTOF)[36] at 3 days after treatment with isotype control, anti-PD-1, RT, or anti-PD-1 and RT. Using CyTOF, we found that PD-L1+ macrophages were more abundant in transplant tumors at baseline and decreased after treatment with RT (Fig. 5b), suggesting a possible mechanism contributing to transplant tumor response to PD-1 blockade and RT. By contrast, PD-L1+ macrophages were relatively rare in primary tumors and did not change significantly after treatment with RT.

To gain insight into the transcriptional differences in the immune microenvironments of primary and transplant tumors, we performed single-cell RNA sequencing (scRNA-seq) on FACS-sorted CD45+ tumor-infiltrating immune cells from sarcomas harvested 3 days after treatment with either anti-PD-1 antibody or isotype control (primary and transplant) and 0 or 20 Gy (primary tumors only). After filtering and quality control, scRNA-seq analysis yielded data for 98,219 cells with 52,220

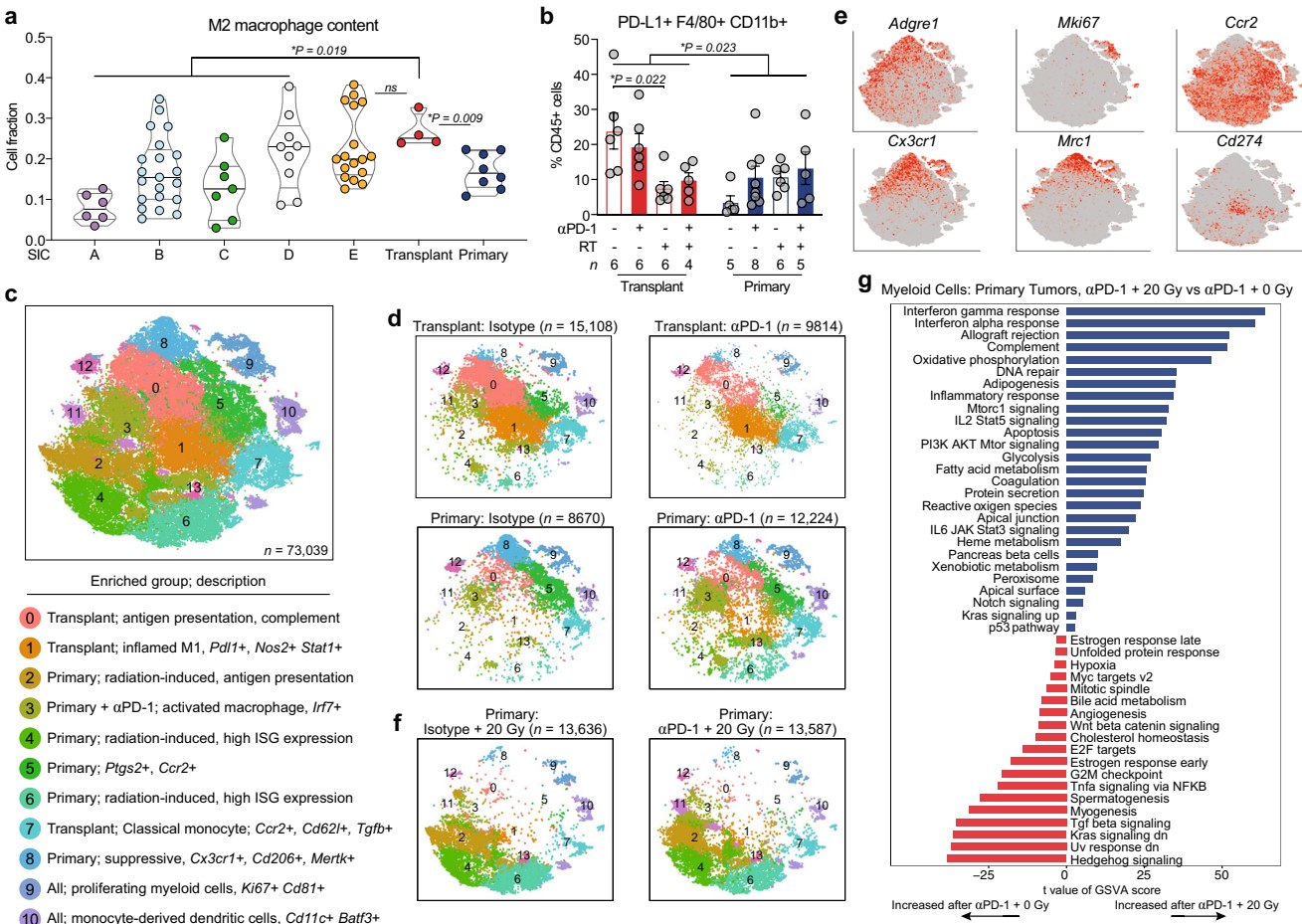

**Fig. 5 Tumor and treatment promote myeloid cell remodeling. a** M2 macrophage content of TCGA UPS (by SIC) and murine sarcomas by CIBERSORTx. *P* values shown for SIC A–D, E, and primary tumors each vs transplant tumors by two-sided Wilcoxon test. Sample sizes as follows: SIC A (*n* = 6), SIC B (*n* = 21), SIC C (*n* = 7), SIC D (*n* = 9), SIC E (*n* = 17), primary (*n* = 8), and transplant (*n* = 4). **b** Frequency of PD-L1+ macrophages by CyTOF. Data show mean ± SEM, analyzed by three-way ANOVA. **c** tSNE plot of myeloid cell scRNA-seq subclustering from all tumor and treatment groups. For major clusters, legend lists tumor/treatment group(s) where each cluster is enriched and description. ISG, interferon-stimulated gene. **d** Distribution of myeloid cells, separated by tumor and treatment type. *n* = 4 tumors per group for transplant, *n* = 5 tumors per group for primary. **e** Expression levels of marker genes. **f** Distribution of myeloid cells from primary tumors treated with 20 Gy RT and isotype (left) or αPD-1 (*n* = 5 per group). **g** Pathways with significantly different activities per cell using GSVA between myeloid cells from primary tumors treated with αPD-1 and 20 Gy vs αPD-1 and 0 Gy (*n* = 25,811 cells analyzed). Pathways shown are significant at false-discovery rate <0.01. Source data are provided as a Source data file.

mean reads per cell, detecting 1570 median genes per cell. We performed tSNE dimensionality reduction and graph-based clustering of all cells in aggregate to identify cells with distinct transcriptional profiles (Supplementary Fig. 4a, b). Graph-based clustering identified 23 cell clusters with distinct transcriptional programs that could be readily assigned to known cell lineages using marker genes (Supplementary Fig. 4c), resulting in classification of two B cell clusters (C19, C22), nine myeloid cell clusters (C0, C1, C2, C3, C4, C6, C8, C11, and C18), six T/natural killer (NK) cell clusters (C5, C7, C9, C12, C13, and C16), two conventional dendritic cells (cDC) (C20, C21) clusters, one cluster of plasmacytoid dendritic cells (pDC) (C14), one cluster of neutrophils (C10), one cluster of osteoblasts (C15), and one cluster of fibroblasts (C17), which was subsequently eliminated from the analysis. Single-cell profiling revealed that multiple immune cell populations were enriched specifically in either primary or transplant sarcomas (Supplementary Fig. 4c, d). For example, myeloid cell cluster C3 was prominent only in transplant tumors, while cluster C6 was enriched in primary tumors (Supplementary Fig. 4b), reflecting the distinct transcriptional states within the myeloid cell compartments of each model.

Furthermore, these scRNA-seq data showed marked differences by tumor model in tumor-infiltrating immune cells based on transcriptional profile and population distribution, which were accentuated by remodeling after anti-PD-1 therapy.

**Tumor and treatment promote myeloid cell remodeling.** Tumor-infiltrating myeloid cells comprise the largest fraction of immune cells in both primary and transplant tumors, and they undergo significant remodeling with anti-PD-1 therapy (Supplementary Fig. 4a–d). To interrogate transcriptional differences in the myeloid phenotypes of primary and transplant sarcomas, we sub-clustered the 73,039 myeloid cells identified by scRNA-seq. This analysis yielded 14 myeloid subpopulations (Fig. 5c, d, and Supplementary Fig. 5a–c), which we compared to published data sets for cell-type identification (Supplementary Fig. 5b). Enrichment of certain myeloid cell clusters specifically in primary or transplant tumors (Fig. 5d, Supplementary Fig. 5a) reflects the transcriptional differences within the myeloid cell compartments of these models, while the incomplete separation between clusters suggests overlapping and highly plastic myeloid cell phenotypes.

The vast majority of myeloid cells from untreated (isotype) transplant tumors fell into clusters Y0 and Y1, which were dominated by proinflammatory genes related to complement (*C1qc, C1qa,* and *C1qb*), interferon-stimulated cytokines (*Cxcl14, Cxcl16, Ccl3,* and *Ccl4*), and transcription factors (*Stat1* and *Irf1*) that cooperate to increase transcription of *Cd274* (PD-L1)[37] (Fig. 5b–e, Supplementary Fig. 5a, c, Supplementary Data S2, S3). A smaller portion of cells from transplant tumors (clusters Y5 and Y7) expressed immunosuppressive genes including *Ptgs2, Ccr2, Chil3,* and *Tgfb*, consistent with M2 macrophage phenotypes (Fig. 5c, d, Supplementary Fig. 5a, c, Supplementary Data S2, S3).

In contrast, the majority of myeloid cells (>60%) from untreated primary tumors represented immunosuppressive myeloid populations (clusters Y5, Y7, and Y8) (Fig. 5c, d, Supplementary Fig. 5a), each which had high expression of a subset of the anti-inflammatory genes *Mrc1, Cx3cr1, Ptgs2,* and *Mertk* (Supplementary Fig. 5c). Interestingly, macrophages found in normal limb muscle from two published data sets[38,39] were most similar to the cells in cluster Y8 (Supplementary Fig. 5d), which was most abundant in untreated primary tumors (Supplementary Fig. 5a). Proinflammatory myeloid cells in primary tumors fell into cluster Y3, which expressed genes associated with antigen presentation and classical macrophage activation (*Cd74, Ccr2, H2-Aa,* and *Cd72*). Although Y3 myeloid cells in primary tumors lacked the *Stat1/Irf1* gene expression signature seen in the proinflammatory myeloid cells from transplants, they did express high levels of *Irf7* (Supplementary Fig. 5c and Supplementary Data 2), a crucial regulator of the type I interferon response and monocyte-to-macrophage differentiation[40]. Taken together, these data illustrate the differences in myeloid phenotypes between untreated primary and transplant tumors: in primary tumors, the majority of myeloid cells express markers associated with resistance to radiation and immunotherapy[41–44], while in transplant tumors, myeloid cells exhibit an interferon-dominated expression signature associated with antitumor immunity[18].

Upon treatment with anti-PD-1 therapy, myeloid cells in transplant tumors further upregulated *Stat1, Irf1,* and *Cd274* (PD-L1), which was accompanied by transcriptional downregulation of the anti-inflammatory macrophage marker *Cx3cr1* (Supplementary Fig. 5c and Supplementary Data 2). These findings support a model where the transcriptional response to anti-PD-1 therapy in transplant-infiltrating myeloid cells is dominated by interferon gamma[18]. In primary tumors, treatment with anti-PD-1 therapy also upregulated genes involved in the type I interferon response (*Irf7, Isg15, Ifit1, Ifit3,* and *Ccl5*) and antigen processing machinery (*Tap1, Tapbp,* and *B2m*) (Supplementary Fig. 5c and Supplementary Data 2), while downregulating immunosuppressive macrophage markers, including *Mrc1* and *Ptgs2*. PD-1 blockade also induced *Stat1* and *Irf1*, suggesting that despite the immunosuppressive myeloid cell environment in isotype control-treated primary tumors, treatment with PD-1 blockade can induce myeloid cells to adopt an antitumor phenotype.

In preclinical studies using transplanted tumor models, focal RT can synergize with immune checkpoint inhibitors by increasing tumor immunogenicity and by reinvigorating the antitumor immune response[9,12,45]. To examine the transcriptional effects of RT on the immune microenvironment of radiation-resistant primary tumors, we also performed scRNA-seq on CD45+ cells isolated from primary sarcomas harvested 3 days after treatment with 20 Gy RT and either anti-PD-1 or isotype control antibody. Compared to unirradiated primary sarcomas, RT dramatically reshaped the transcriptome and distribution of myeloid cell clusters (Fig. 5d, f).

After RT, myeloid cells from primary tumors treated with isotype control or anti-PD-1 antibody clustered into myeloid subclusters Y2, Y4, and Y6 (Fig. 5f and Supplementary Fig. 5e). Y2 macrophages expressed high levels of *Ccl2* and *Ly6C* (Supplementary Fig. 5f) and also expressed genes consistent with active phagocytosis and antibody-dependent cell-mediated cytotoxicity (*Fcgr2b, Fcgr1,* and *Fcgr3*) and antigen/protein processing (*B2m, Ctsl, Ctsd, Ctsb,* and *Ctss*) (Supplementary Data 2). Clusters Y4 and Y6 expressed high levels of the transcription factors *Stat1, Stat2, Irf5,* and *Irf7*, which were likely responsible for the increased expression of interferon-related genes (*Mx1, Ifit1, Ifit3, Ifit3b,* and *Ifit2*) and genes involved in antigen processing and presentation (*Tapbp, Tap1,* and *Scimp*) (Supplementary Fig. 5f and Supplementary Data 2). Compared to anti-PD-1 antibody alone, the addition of RT to treatment of primary tumors increased activity of interferon response pathways and decreased activity of TGF-β signaling in myeloid cells (Fig. 5g). Notably, TGF-β signaling has been associated with resistance to both immune checkpoint blockade and radiotherapy-induced antitumor immunity[46,47]. Taken together, these data indicate that, despite the immunosuppressive microenvironment of untreated primary tumors, PD-1 blockade and RT successfully repolarize myeloid cells in primary tumors, with the dominant changes being activation of type I and II interferon response pathways. However, reprogramming myeloid cells by this combination therapy remains insufficient to cure the primary p53/MCA tumors.

**CD8+ T cell signatures in primary and transplant sarcomas**. Tumor-infiltrating CD8+ T cells have been shown to correlate with the likelihood of response to anti-PD-1 therapy in many tumor types, including soft tissue sarcoma[20,48]. To examine how the CD8+ T cell abundance in primary and transplant tumors compared to the various sarcoma immune classes in patient tumors, we performed CIBERSORTx analysis on human TCGA data and untreated primary and transplant murine p53/MCA sarcomas. In bulk tumor gene expression data, the transplant tumor CD8+ T cell abundance mirrored that of the human immune-high SIC E tumors, while primary tumors were more similar to the immune-low human sarcomas (Fig. 6a).

CyTOF profiling revealed that a subset of CD8+ T cells within transplant tumors expressed the immune checkpoint molecules Lag3 and Tim3 (Fig. 6b), which are upregulated upon T cell activation[49]. In contrast, fewer CD8+ T cells expressing these activation markers were present in primary tumors, demonstrating that primary and transplant tumors promote distinct phenotypes within tumor-infiltrating CD8+ T cells. To test whether CD8+ T cells were necessary for transplant tumor cure by PD-1 blockade and RT, we depleted CD8+ T cells before and during treatment. We found that depletion of CD8+ T cells abrogated the effects of PD-1 blockade, RT, and combination treatment in transplant tumors (Fig. 6c). Interestingly, we also found that transplant tumor cure by anti-PD-1 and RT was dependent on CD4+ T cells (Fig. 6d).

To examine the lymphoid populations identified by scRNA-seq, we computationally separated 14,705 lymphoid cells from all CD45+ cells for further analysis (Fig. 6e). This approach yielded 12 distinct subpopulations, including regulatory T cells (Treg) (L5), naive CD4+ T cells (L9), CD8+ T cells (L0, L4, L6, and L8), one population containing both CD4+ and CD8+ memory T cells (L3), natural killer cells (L1, L10), B cells (L7), plasma cells (L12), and one myeloid population which was subsequently removed from analysis (L11) (Fig. 6e, f, Supplementary Fig. 6).

The most prominent difference was the low number of activated CD8+ T cells (L0) in primary tumors relative to transplant tumors

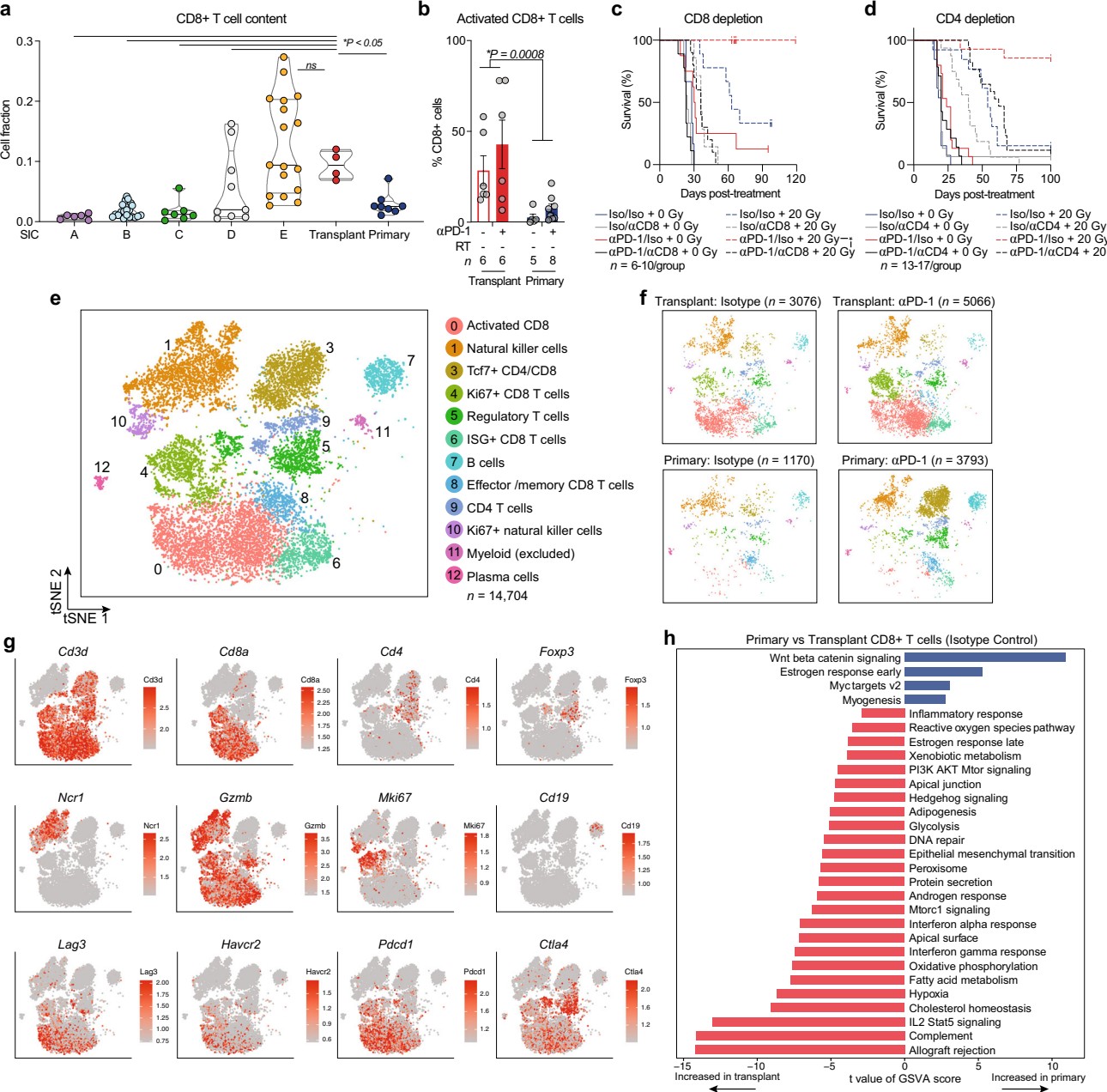

**Fig. 6 Tumor and treatment promote lymphoid cell remodeling. a** CD8+ T cell content of TCGA UPS by sarcoma immune class (SIC) and murine sarcomas by CIBERSORTx. *P* values shown for each group vs transplant tumors by two-sided Wilcoxon test. Sample sizes as follows: SIC A ($n = 6$), SIC B ($n = 21$), SIC C ($n = 7$), SIC D ($n = 9$), SIC E ($n = 17$), primary ($n = 8$), and transplant ($n = 4$). **b** Frequency of activated (Lag3+ Tim3+) CD8+ T cells by CyTOF. Data show mean ± SEM by two-way ANOVA. **c** Mice with transplant sarcomas received either αPD-1 or isotype control antibodies, αCD8 or isotype control, and 0 Gy (solid) or 20 Gy (dashed) when tumors reached >70 mm³. Surviving mice were censored 164 days after treatment (*$P = 0.002$). Sample sizes: Iso/Iso + 0 Gy $n = 6$, Iso/αCD8 + 0 Gy $n = 9$, αPD-1/Iso + 0 Gy $n = 8$, αPD-1/αCD8 + 0 Gy $n = 9$, Iso/Iso + 20 Gy $n = 9$, Iso/αCD8 + 20 Gy $n = 7$, αPD-1/Iso + 20 Gy $n = 7$, αPD-1/αCD8 + 20 Gy $n = 10$. **d** Mice with transplant sarcomas received either αPD-1 or isotype control antibodies, αCD4 or isotype control, and 0 Gy (solid) or 20 Gy (dashed) when tumors reached >70 mm³. Surviving mice were censored 100 days after treatment (*$P = 0.002$). Sample sizes: Iso/Iso + 0 Gy $n = 13$, Iso/αCD4 + 0 Gy $n = 15$, αPD-1/Iso + 0 Gy $n = 15$, αPD-1/αCD4 + 0 Gy $n = 14$, Iso/Iso + 20 Gy $n = 13$, Iso/αCD4 + 20 Gy $n = 16$, αPD-1/Iso + 20 Gy $n = 14$, αPD-1/αCD4 + 20 Gy $n = 17$. **e** tSNE plot of lymphoid cell scRNA-seq subclustering from all tumor and treatment groups. **f** Distribution of lymphoid cells, separated by tumor and treatment type. $n = 4$ tumors per group for transplant, $n = 5$ tumors per group for primary. **g** Expression levels of marker genes. **h** Pathways with significantly different activities per cell using GSVA between CD8+ T cells from untreated (isotype control) primary tumors versus transplant tumors ($n = 2473$ cells analyzed). Pathways shown are significant at false-discovery rate <0.01. For **c** and **d**, survival curves estimated using Kaplan–Meier method; pairwise significance determined by log-rank test and Bonferroni correction. Source data are provided as a Source data file.

(Fig. 6e–g, Supplementary Fig. 6a, and Supplementary Data 3). In contrast, the majority of CD8+ T cells from transplant tumors fell into population L0, which expressed high levels of genes associated with T cell activation including *Pdcd1, Havcr2, Lag3, Ctla4, Cd38,* and *Entpd1* (Fig. 6e–g, Supplementary Fig. 6a, b, and Supplementary Data 4). In both cancer and chronic viral infection, a subset of T cells expressing high levels of PD-1 (*Pdcd1*) and other inhibitory checkpoints proliferates and differentiates into effectors that can mediate long-term immune control after anti-PD-1 immunotherapy[50,51]. This suggests that activated CD8+ T cells in cluster L0, which express the immune inhibitory checkpoint molecules Tim3 (*Havcr2*) and Lag3, and which are enriched in transplant but not primary tumors, may mediate transplant tumor cure by anti-PD-1 therapy. Interestingly, within the small number of CD8+ T cells from primary tumors that fell into cluster L0, treatment with anti-PD-1 antibody induced high expression of *Tox* (Supplementary Fig. 6b and Supplementary Data 4), a critical regulator of tumor-specific T cell differentiation that promotes T cell commitment to an exhausted and dysfunctional phenotype[52–54].

Within the CD8+ T cell populations, L0, L4, and L6 were enriched in transplant tumors (Fig. 6f and Supplementary Fig. 6a). Actively cycling CD8+ T cells (*Cdk1, Ccnb1, Ccna2, Mki67,* and *Cdk4*) within population L4 were also more abundant within transplant tumors and, consistent with previous studies[18], increased after anti-PD-1 treatment (Fig. 6f, g, Supplementary Fig. 6a, and Supplementary Data 4). Multiplex immunofluorescence staining verified these findings, demonstrating that there were significantly more activated Lag3+ CD8+ T cells (corresponding to cluster L0) and cycling Ki67+ CD8+ T cells (corresponding to cluster L4) in transplant tumors compared to primary tumors after anti-PD-1 therapy (Supplementary Fig. 6c, d). Population L6 expressed high levels of genes associated with the type I interferon response (*Stat1, Irf7, Mx1, Ifit1, Isg15,* and *Ccl5*) and moderate levels of the same activation markers found in L0 (Supplementary Fig. 6b and Supplementary Data 4). L8, a memory population which lacked exhaustion markers and primarily expressed markers associated with survival/proliferation (*Il7r, Slamf6, Ccnd3, Ccnd2,* and *Cd69*), was present at a similar frequency in primary and transplant tumors (Supplementary Fig. 6b and Supplementary Data 4).

Differential expression analysis of all CD8+ T cells from untreated tumors revealed that CD8+ T cells in transplant tumors exhibited higher levels of activation/exhaustion markers (*Gzmb, Lag3, Havcr2, Tnfrsf9, Icos,* and *Pdcd1*) and increased activity of transplant rejection and interferon response pathways (Fig. 6h and Supplementary Data 4). Comparison of pathway activity in CD8+ T cells revealed elevated Wnt/β-Catenin and Myc target gene signaling in primary tumors compared to transplant tumors at baseline (isotype control treatment) (Fig. 6h). Within CD8+ T cells from transplant tumors, treatment with anti-PD-1 antibody increased expression of granzymes (*Gzma, Gzmb*) and cell proliferation genes (*Rps12, Rpl5, Eif4a1,* and *Top2a*), while anti-PD-1 treatment reduced expression of the exhaustion markers *Tnfrsf18* (*Gitr*) and *Lag3* (Supplementary Fig. 6b and Supplementary Data 4). Within CD8+ T cells in primary tumors, anti-PD-1 therapy had similar effects on granzyme expression (increased *Gzma, Gzmb*), proliferation genes (increased *Rps12, Rpl5,* and *Smchd1*), and exhaustion markers (decreased *Tnfrsf18* and *Pdcd1*) (Supplementary Fig. 6b and Supplementary Data 4). Interestingly, the activity of the Wnt/β-Catenin signaling pathway in CD8+ T cells increased further after anti-PD-1 treatment in primary tumors but decreased after anti-PD-1 treatment in transplant tumors (Supplementary Fig. 7a, b). Lymphocyte overexpression of Wnt/β-Catenin target genes has been shown to lead to apoptosis in mature T cells[55], and tumor cell-intrinsic β-Catenin signaling is

associated with T cell exclusion and immune evasion[56]. These findings suggest that the anti-PD-1 antibody was effectively engaging T cells in both models, but that transplant and primary tumors promote disparate CD8+ T cell states that are associated with response or resistance to anti-PD-1 immunotherapy.

## Discussion

Here, we have demonstrated that primary and transplant sarcomas derived from the same tumor model have distinct immune landscapes with divergent responses to radiotherapy and anti-PD-1 immunotherapy. Although transplant tumors upregulated expression of tumor neoantigens, tumor transplantation alone was insufficient to sensitize tumors to PD-1 blockade and radiation therapy. By generating autochthonous tumors in *Rag2*$^{−/−}$ immunodeficient and *Rag2*$^{+/−}$ immunocompetent mice, and by performing complementary self vs. non-self tumor transplantation experiments, we observed that this therapeutic resistance was driven by reciprocal immunoediting and suppression of neoantigen expression in the tumor cells and by reshaping of the immune system during the development and progression of primary tumors. We identified distinct cellular ecosystems in primary and transplant tumor models and compared them to the composition of human sarcomas. We found that untreated transplant tumors from mice were enriched for CD8+ T cells and M2 macrophages, closely resembling the highly inflamed human sarcomas that are most likely to respond to PD-1 checkpoint blockade. By contrast, primary tumors resembled the majority of human sarcomas, which are immune-low and unlikely to respond to PD-1 blockade[19]. By single-cell sequencing and mass cytometry, we revealed distinct myeloid and T cell phenotypes that were specific to either primary or transplant tumors. Our observations suggest that primary tumors, which coevolve with their immune system, recapitulate the immune-low microenvironment and resistance to PD-1 blockade seen in the majority of patient sarcomas. In contrast, transplant tumors promote an immune landscape that favors antitumor immunity, similar to the small fraction of highly inflamed patient sarcomas (SIC E) that respond to PD-1 blockade. Although B cells have been shown to be correlated with survival and immunotherapy response in patients with sarcomas[19], we found that immunotherapy-responsive transplant tumors with high levels of inflammation had fewer B cells than immunotherapy-resistant primary tumors.

Because primary and transplant models of cancer have distinct immune landscapes, they may rely on distinct mechanisms for immunologic clearance. For example, PD-1 blockade in transplant tumors induces a decrease in immunosuppressive M2 macrophages, which is accompanied by an increase in activated CD8+ T cells. Similar proinflammatory myeloid cell remodeling occurs in primary tumors after PD-1 blockade, but this is not accompanied by an increase in activated CD8+ T cells in primary tumors, suggesting that additional mechanisms of immune tolerance are responsible for primary tumor resistance to immunotherapy. Although the responses in transplant tumors suggest synergy between anti-PD-1 therapy and RT, we do not identify synergistic effects to improve survival when mice with primary tumors are treated with this combination therapy. Nevertheless, immune checkpoint blockade therapy alone has activity in a subset of cancer patients. Therefore, even if ongoing clinical trials reveal that there is no synergy between radiation therapy and checkpoint blockade in patients, there may be cooperative effects which have the potential to improve patient outcomes if resistance mechanisms can be overcome.

Immune-high tumors may exhibit unique mechanisms of resistance to immunotherapy and radiation therapy, which has

important implications for interpreting preclinical data and translating them to clinical trials. After treatment with anti-PD-1 therapy, transplant but not primary tumors upregulate *Ido1* (Supplementary Fig. 3c). Although Ido1 inhibitors enhance the efficacy of anti-PD-1 therapy in transplant mouse models[11], this combination failed to benefit patients with melanoma[57]. Generating primary tumors that capture the heterogeneity of human tumors and testing their responses to therapy requires a significant investment of time and resources in comparison to studying transplant tumors. However, the effort and resources required to perform these studies in primary tumor models are much less compared to those required for clinical trials, which also carry risks for human subjects. Results of cancer therapies in transplant tumor models often fail to predict efficacy for cancer patients[58]. The finding that many cancer therapeutics with promising preclinical results fail to demonstrate efficacy in clinical trials suggests that the transplant models commonly employed for preclinical studies do not fully recapitulate the complex nature and diversity of human cancers, which include their interactions with host immune responses, intratumoral heterogeneity, and the diverse cell types present within the tumor microenvironment. Our findings suggest that transplant tumor models may recapitulate the immune microenvironment of highly inflamed tumor subtypes that are likely to respond to immunotherapy, but do not resemble the majority of human cancers, which are resistant to immunotherapy. Performing complementary studies in transplant and autochthonous mouse models may thus increase the success of translation of immunotherapies from preclinical studies to patients.

## Methods

**Mouse strains.** All animal studies were performed in accordance with protocols approved by the Duke University Institutional Animal Care and Use Committee (IACUC) and adhere to the NIH Guide for the Care and Use of Laboratory Animals. The *Trp53^{fl/fl}* allele used in this study has been described previously[59]. *Trp53^{fl/fl}* and wild-type mice were maintained on a pure 129/SvJae genetic background and bred at Duke University. Nude (*nu/nu*) mice were purchased from Taconic. To minimize the effects of sex and genetic background, male and female mice and age-matched littermate controls were used for every experiment so that potential genetic modifiers would be randomly distributed between experimental and control groups.

**Sarcoma induction and treatment.** Primary p53/MCA sarcomas were generated in 129/SvJae mice between 6 and 10 weeks old by intramuscular injection of adenovirus expressing Cre recombinase (Adeno-Cre; University of Iowa Viral Vector Core) in *Trp53^{fl/fl}* mice. Twenty-five microliters of adenovirus was mixed with 600 μL DMEM (Gibco) and 3 μL 2 M CaCl₂, then incubated for 15 min at room temperature prior to injection. Fifty microliters of the prepared mixture was injected into the gastrocnemius muscle of the mice, followed by injection of 300 μg MCA (Sigma) resuspended in sesame oil (Sigma) at 6 μg/μL. Transplant p53/MCA sarcomas were generated by injecting 50,000 cells resuspended in 100 μL of a 1:1 mixture of DMEM (Gibco) and Matrigel (Corning) into the gastrocnemius muscle.

Primary p53/MCA sarcomas were generated in *Rag2^{−/−}* and *Rag2^{+/−}* mice by intramuscular injection of adenovirus expressing the sgRNA targeting *Trp53* (sgp53) and Cas9 endonuclease (Adeno-sgp53-Cas9; Viraquest), sgp53 sequence: GATGGTAAGGATAGGTCGG. Twenty-five microliters of Adeno-sgp53-Cas9 was mixed with 600 μL DMEM (Gibco) and 3 μL 2 M CaCl₂, then incubated for 15 min at room temperature prior to injection. Fifty microliters of the prepared mixture was injected into the gastrocnemius muscle of each mouse, followed by injection of 300 μg MCA (Sigma) resuspended in sesame oil (Sigma) at 6 μg/μL. Tumors were collected when they reached 70–150 mm³.

When tumors reached 70–150 mm³ (Day 0, D0), 129/SvJae mice were randomized to treatment groups, then tumors were monitored three times weekly by caliper measurements in two dimensions. Antibodies were administered starting on D0 (prior to radiation treatment) by intraperitoneal injection of 200 μL per dose at 1 mg/mL diluted in PBS. Anti-PD-1 (MSD muDX400 in all figures except Fig. 1c, d with BioXCell, BE0146) and anti-CTLA-4 (BioXCell, BE0164) or isotype control (MSD IgG1 control for muDX400, BioXCell, BE0086 for CTLA-4) treatments were administered on days 0, 3, and 6. Anti-CD4 (BioXCell, BE0003-1) and anti-CD8 (BioXCell, BE0061) or isotype control (BioXCell, BE0090) antibodies were injected on day 0, 3, 6, followed by weekly injections for the rest of the experiment. Samples shown in Fig. 1a are a subset of those shown in Fig. 6c (non-CD8-depleted animals only). Figure 6d shows the combined results from two independent experiments.

Mice were considered "cured" and censored on Kaplan– Meier analysis if tumors were undetectable by caliper measurement for at least 60 days.

Sarcoma irradiations were performed using the Precision Xrad 225Cx small animal image-guided irradiator[60]. The irradiation field was centered on the target via fluoroscopy with 40 kilovolt peak (kVp), 2.5 mA X-rays using a 0.3 mm copper filter. Sarcomas were irradiated with parallel-opposed anterior and posterior fields with an average dose rate of 300 cGy/min prescribed to midplane at treatment isocenter with 225 kVp, 13 mA X-rays using a 0.3 mm copper filter, and a collimator with a 40 × 40 mm² radiation field.

Mice were euthanized with CO₂ if moribund or when tumor volumes reached more than 13 mm in any dimension, in accordance with IACUC guidelines at Duke University.

**Tumor transplantation experiments.** For amputation and tumor transplantation experiments, primary tumors were generated using Adeno-Cre and MCA as described above. After hind limb amputation, tumors were dissected from the limb and dissociated by shaking for 45 min at 37 °C in collagenase Type IV (Gibco), dispase (Gibco), and trypsin (Gibco). Cell suspension was then strained through a 40 μm filter, washed in PBS, and plated for culture. Cell lines were maintained in vitro for 10 passages before transplanting into naive or donor mice. For each donor mouse that was transplanted with a "self" cell line, ~5 matched naive mice also received an injection of the same cell line. For both donor and naive mice, transplant sarcomas were generated by injecting 50,000 cells resuspended in 100 μL of a 1:1 mixture of DMEM (Gibco) and Matrigel (Corning) into the gastrocnemius muscle.

*Bone marrow transplant.* Mice receiving bone marrow transplant were treated with 2 fractions of 5.5 Gy total body irradiation (TBI) delivered 18 h apart. TBI was delivered using the Precision Xrad 225Cx small animal image-guided irradiator with no collimator and parallel-opposed anterior and posterior fields. Bone marrow transplant was performed within 3 h of the second TBI dose. Whole bone marrow cells were isolated from femurs and tibias of healthy mice on a 129/SvJae genetic background by washing the bone marrow space with PBS. RBCs were lysed using ACK lysing buffer (Lonza). Bone marrow cells were counted with an automated cell counter (Cellometer Auto 2000, Nexcelom Bioscience) using AO/PI stain (Nexcelom Bioscience). Three million whole bone marrow cells were resuspended in 50 μL PBS and injected retro-orbitally into recipient mice.

**DNA and RNA sequencing of primary and transplant sarcomas in 129/SvJae mice**
*RNA sequencing.* Tumor specimens and matched liver control were harvested and stored in RNALater (Ambion) at −80 °C until all samples were collected. DNA and RNA extractions from each sample were performed using AllPrep DNA/RNA Mini Kit (Qiagen). Extracted total RNA quality and concentration was assessed on a 2100 Bioanalyzer (Agilent Technologies) and Qubit 2.0 (ThermoFisher Scientific), respectively. Only extracts with RNA integrity number >7 were processed for sequencing. RNA-seq libraries were prepared using the KAPA Stranded mRNA-Seq Kit (Roche) following the manufacturer's protocol. mRNA transcripts were first captured using magnetic oligo-dT beads, fragmented using heat and magnesium, and reverse transcribed using random priming. During the second strand synthesis, the cDNA:RNA hybrid was converted to double-stranded cDNA (dscDNA) and dUTP was incorporated into the second cDNA strand to mark it. Illumina sequencing adapters were then ligated to the dscDNA fragments and amplified to produce the final RNA-seq library. The strand marked with dUTP was not amplified, allowing strand-specific sequencing. Libraries were indexed using a dual indexing approach so that multiple libraries could be pooled and sequenced on the same sequencing Illumina sequencing flow cell. RNA samples were pooled with whole-exome sequencing (WES) libraries for sequencing (see WES section below for additional details). For the RNA samples, sequencing generated an average of ~95 million reads per tumor sample.

*Whole-exome sequencing.* DNA extraction was followed by RNAse treatment (Qiagen). Genomic DNA samples were quantified using fluorometric quantitation on the Qubit 2.0 (ThermoFisher Scientific). For each sample, 200 ng DNA was sheared using focused-ultrasonicators (Covaris) to generate DNA fragments of about 300 bp in length. Sequencing libraries were then prepared using the Agilent SureSelect XT Mouse All Exon kit (#S0276129). During adapter ligation, unique indexes were added to each sample. Resulting libraries were cleaned using solid-phase reversible immobilization beads and quantified on the Qubit 2.0, and size distribution was checked on an Agilent Bioanalyzer. Libraries were subsequently enriched individually by hybridization of the prepared gDNA libraries with mouse all exome target-specific probes provided with the SureSelect XT Mouse All Exon kit (target size 49.6 Mb). After hybridization, the targeted molecules were captured on streptavidin beads. Once enriched, the captured libraries were pooled with the RNA libraries and sequenced on an Illumina Novaseq 6000 S4 flow cell at 151 bp paired-end. Base-calling was done on the instrument using RTA v3.3.3. For WES samples, sequencing generated an average of ~150 million reads per tumor sample, with tumor coverage at an average depth of 625X, and an average of ~38 million reads per liver sample, with liver coverage at an average depth of 157X. Once

generated, sequence data were demultiplexed and Fastq files were generated using Bcl2Fastq2 conversion software provided by Illumina (v2.20.0.422).

**DNA and RNA isolation and sequencing from p53/MCA sarcomas in Rag2⁻/⁻ and Rag2⁺/⁻ mice.** Tumor tissue was collected and isolated from tumors as described above. Tail was used for matched normal tissue control, and tail DNA was extracted using the Qiagen DNeasy Blood and Tissue Kit. RNA Library preparation was performed as indicated by the NEBNext Ultra II RNA Library Prep Kit for Illumina, and RNA sequencing was performed by Novogene on an Illumina Novoseq 6000 in 150 base pair, paired-end configuration. WES libraries were prepared as indicated by the Agilent SureSelect^XT2 Mouse All Exon V6 Kit, and WES was performed by Novogene on an Illumina Novoseq 6000 in 150 base pair, paired-end configuration. Samples that failed QC or had insufficient total amount of RNA or DNA were subsequently removed from analysis.

**RNA sequencing analysis for differential expression**
*Preprocessing.* The quality of the sequencing reads was first assessed using FastQC (v0.11.5)[61] and MultiQC[62]. Low quality reads and adapters were detected and removed with Trimmomatic (v0.36)[63]. The quality of the reads was assessed again before downstream analyses and qualified reads were then aligned to UCSC mm10 mouse genome from the iGenome project using STAR (v2.5.4b)[64] and mapped to the mouse transcriptome annotated by GENCODE (Release M17)[65]. Summaries for alignment and mapping performance are provided in Supplementary Data 5. Raw gene counts were quantified using HTSeq[66] implemented in the STAR pipeline.

*Gene differential expression analysis.* Normalization of gene counts and differential expression analysis were performed based on modeling the raw counts within the framework of a negative binomial model using the R package DESeq2 (v1.20.0)[67]. Pathway analyses, based on Gene Ontology (GO) terms were conducted using the gage package (v2.34.0)[68]. The Benjamini–Hochberg method[69] was used to adjust $p$ values for multiple testing within the false-discovery framework.

**CIBERSORTx RNA sequencing analysis.** For TCGA analyses, gene expression levels were summarized as transcripts per million then used as input for CIBERSORTx deconvolution with the LM22 and a modified TM4 signature matrices with B-mode batch correction using the web application for CIBERSORTx (cibersortx. stanford.edu)[31]. Since sarcomas do not contain epithelial cells, this subpopulation was removed from the TM4 signature matrix to create a "TM3" signature matrix. Similarly, for bulk mouse tumor transcriptomes, the LM22 and TM3 signature matrices were mapped to orthologous mouse genes and used for deconvolution by CIBERSORTx. The TM3 signature matrix was used to quantify immune and non-immune cell proportions, and the LM22 signature matrix enumerated immune cell proportions. For downstream analyses, the immune cell subtype proportions from LM22 were scaled to match the total immune cell proportion from TM3, which resulted in relative quantification of immune cell subtypes as fraction of all cells in each respective tumor. The Wilcoxon test was used to compare groups as indicated in the figure legends. For the heatmap presentation, these values were standardized as $z$-scores. All analyses were conducted using R (v.3.5.1) using base R and/or the ggplot2 and ComplexHeatmap packages.

**RNA sequencing analysis for neoantigen expression.** Alignment of RNA sequencing data was performed in Omicsoft Array Studio (v10.0.1.118). Briefly, cleaned reads were aligned to the mouse B38 genome reference by using the Omicsoft Sequence Aligner (OSA)[70], with a maximum of two allowed mismatches. Gene level counts were determined by the OSA algorithm as implemented in Omicsoft Array Studio and using Ensembl.R86 gene models. Approximately 88% of reads across all samples mapped to the mouse genome (corresponding to ~84 million reads on average).

**Whole-exome sequencing analysis**
*Somatic mutation calling.* The WES reads were aligned to the mouse reference genome mm10 using the BWA-MEM algorithm (v0.7.12)[71]. The reference genome was obtained from the UCSC FTP site (ftp://hgdownload.soe.ucsc.edu/goldenPath/mm10/). The original Agilent bait designed BED file, which had been built on GRCm37 (mm9), was lifted over to GRCm38 (mm10) to match the other reference files.

The aligned bam files were post-processed by following the recommended pipeline of Genome Analysis Toolkit (GATK, version 3.8)[72] to generate the analysis-ready BAM files for variant calling. Briefly, first, Picard (v1.114; http://broadinstitute.github.io/picard/faq.html) MarkDuplicates module was used for identifying PCR duplication. Afterward, the reads were locally realigned around insertion or deletion (indels) by module RealignerTargetCreator/IndelRealigner of GATK. Finally, module BaseRecalibrator of GATK was performed to recalibrate quality scores.

Somatic mutations were detected using GATK3.7 MuTect2[73] with default parameters by inputting the analysis-ready BAM files of tumor and matched normal tissue (liver) control for each animal. Parameter "--dbsnp" was assigned with mouse Single Nucleotide Polymorphism Database (dbSNP) (v150)[74]. Variants called by MuTect2 that were present in the dbSNP were removed. Variants with a mutant allele depth <4 or total read depth <15 were excluded. Variants were annotated with their most deleterious effects on Ensembl transcripts with Ensembl VEP (Variant Effect Predictor, Version 88)[75] on GRCm38. Tumor mutation burden was defined as the number of somatic nonsynonymous variants (including SNVs and indels) that passed the described filters. Mutation data are provided in Supplementary Data 6.

*Neoantigen prediction.* Mutant 8–11mer peptides that could arise from the identified non-silent mutations present in each tumor were identified. If the variant gave rise to a single amino acid change, the mutant peptide was scanned with a sliding window of 8–11 amino acids around the variant to generate all possible 8, 9, 10, and 11mers. If the variant created large novel stretches of amino acids that were not present in the reference genome (e.g., stop losses or frameshifts), all possible peptides of 8, 9, 10, and 11mers were extracted from the large novel peptide. The binding ability between all the mutant peptides and mouse H2-K^b/D^b was predicted by netMHC (4.0)[76] with default parameters. For each specific variant, the associated peptides were considered to be neoantigens if they met the following criteria: the half-maximal inhibitory concentration binding affinity scores (affinity score) of mutant peptides <500 nM and reference peptide affinity score >500 nM. The number of RNA-seq reads covering each of the predicted variants was extracted from the samtools[77] (v1.6) mpileup results (with RNA-Seq bam files processed by Omicsoft (see the section "RNA sequencing analysis for neoantigen expression" above). Neoantigens were considered expressed if both the mutant and the reference alleles were found in at least five RNA-seq reads. Neoantigen expression fraction was calculated by dividing the number of nonsynonymous mutations with expressed neoantigens by the total number of nonsynonymous mutations giving rise to neoantigens. Neoantigen expression level was determined by identifying the fragments per kilobase of transcript per million mapped reads (FPKM) of the corresponding genes, then upper quartile normalized and $\log_2$ transformed.

**Mass cytometry**
*Staining.* Antibody clones and sources are listed in Supplementary Table 1. For custom-conjugated antibodies, 100 μg of antibody was coupled to Maxpar X8 metal-labeled polymer according to the manufacturer's protocol (Fluidigm). After conjugation, the metal-labeled antibodies were diluted in Antibody Stabilizer PBS (Candor Bioscience) for long-term storage. After tumor dissociation and RBC lysis as described above, three million cells per sample were transferred to 5 mL round-bottom tubes (Corning). Cells were incubated with 300 μL of Cell-ID Cisplatin-195Pt (Fluidigm) diluted 1:4000 in Maxpar PBS (Fluidigm) for 5 min at room temperature, then washed twice with Maxpar Cell Staining Buffer (CSB) (Fluidigm). Samples were incubated with 50 μL FcR Blocking Reagent (Biolegend, 1:100 dilution) for 10 min at room temperature, then 50 μL extracellular antibody cocktail was added and incubated for 30 min total at room temperature. Cells were washed twice with CSB, then fixed and permeabilized with eBioscience Foxp3/Transcription Factor Fixation/Permeabilization Buffer for 1 h at room temperature, followed by two washes with permeabilization buffer (eBioscience). Fifty microliters of intracellular antibody cocktail in permeabilization buffer was added and incubated for 30 min total at room temperature, followed by two washes with permeabilization buffer. Cells were fixed in 1.6% methanol-free PFA (Thermofisher) diluted with Maxpar PBS (Fluidigm) for 1 h at 4 °C. Each sample was barcoded with a unique combination of palladium metal barcodes (Fluidigm) for 30 min. Samples were washed twice, combined, and incubated at least overnight in Maxpar Fix and Perm Buffer (Fluidigm) with 62.5 nM Cell-ID Intercalator (Fluidigm) containing ¹⁹¹Ir and ¹⁹³Ir. Before collection, cells were washed once with CSB, once with Cell Acquisition Solution (CAS) (Fluidigm), then filtered and diluted in CAS containing 10% EQ Calibration Beads (Fluidigm) at 0.5 million cells per mL before acquisition on a mass cytometer (Helios).

*CyTOF data analysis.* Mass cytometry data were normalized, concatenated, and debarcoded using Fluidigm CyTOF software (v7.0). Individual samples were gated in Cytobank[78] to exclude beads, debris, dead cells, and doublets for further analysis. For each experimental group (time point and treatment), cells from 5 to 8 tumors per group were manually gated to identify specific populations.

**Single-cell RNA sequencing**
*Tumor harvest and dissociation.* Tumors were dissected from mice, minced, and digested using the Miltenyi Biotec tumor dissociation kit (mouse, tough tumor dissociation protocol) for 40 min at 37 °C. Cells were then strained through a 70 μm filter and washed with FACS buffer (HBSS (Gibco) with 5 mM EDTA (Sigma-Aldrich) and 2.5% fetal bovine serum (Gibco)). Red blood cells were lysed using ACK lysis buffer (Lonza), washed again with FACS buffer, and strained through a 40 μm filter. Cells were then washed and stained for cell sorting.

*Fluorescence-activated cell sorting.* For sorting of CD45+ cells for single-cell RNA sequencing, single-cell suspensions of tumors were blocked with purified rat anti-mouse CD16/CD32 (BD Pharmingen, dilution 1:100) for 10 min at room

temperature then stained with Live/Dead dye (Zombie Aqua, Biolegend) and anti-mouse CD45 (BV605 or APC-Cy7, Biolegend) for 25 min on ice. Live CD45+ cells were isolated for scRNA-seq using an Astrios (Beckman Coulter) sorter and resuspended in PBS with 0.04% BSA at a concentration of 1000 cells/μL for single-cell RNA sequencing.

*Library preparation and sequencing.* scRNA-seq was performed as described[79]. Briefly, single-cell suspensions from sorted live CD45+ cells were loaded on a GemCode Single Cell instrument (10x Genomics) to generate single-cell beads in emulsion and scRNA-seq libraries were prepared using the Chromium Single Cell 3′ Reagent Kits (v2), including Single Cell 3′ Library & Gel Bead Kit v2 (120237), Single Cell 3′ Chip Kit v2 (PN-120236), and i7 Multiplex Kit (120262) (10x Genomics) following the Single Cell 3′ Reagent Kits (v2) User Guide. Single-cell barcoded cDNA libraries were quantified by quantitative PCR (Kappa Biosystems) and sequenced on an Illumina NextSeq 500 (Illumina). Read lengths were 26 bp for read 1, 8 bp for i7 index, and 98 bp for read 2. Cells were sequenced to greater than 50,000 reads per cell as recommended by manufacturer.

*Analysis of scRNA-seq data.* The CellRanger Single Cell Software Suite (v2.1.1) was used to perform sample de-multiplexing, barcode processing, and single-cell 3′ gene counting. Reads were aligned to the mouse (mm10) reference genome. Cells that had less than 200 expressed genes, more than 7000 expressed genes, or >0.05% of mitochondrial genes were excluded from analysis. Graph-based cell clustering, dimensionality reduction, and data visualization were analyzed by the Seurat R package[80] (v2.4). The number of cell clusters is determined using graph-based clustering in Seurat that embeds cells in a K-nearest neighbor graph based on the Euclidean distance in PCA space and Jaccard similarity to iteratively group cells together with similar gene expression patterns. We determined the number of statistically significant principal components to input into the graph-based clustering algorithm using jackStraw[81], but selected cluster resolution according to Seurat[82] recommendations (https://satijalab.org/seurat/v3.0/pbmc3k_tutorial.html) based on the size of the dataset. We used tSNE visualization to confirm appropriate clustering. Differentially expressed transcripts were determined in the Seurat package utilizing a likelihood-ratio test for single-cell gene expression[83]. Graphics were generated using Seurat, ggplot2 R packages, and Graphpad Prism (v8). To identify subclusters within the lymphoid and myeloid cell compartments, we reanalyzed selected cells within lymphoid and myeloid clusters using the Seurat pipeline described above.

*Gene set variation analysis (GSVA).* To identify functionally enriched pathways between tumor types or between treated and untreated tumor in CD8+ T Cells, we used a modified version of previously described methods[84]. First, to assign pathway activity estimates to individual cells, we applied GSVA[85] to calculate enrichment scores in each cell for the 50 hallmark pathways described in the molecular signature database[86] (MSigDB v6.0), as implemented in the GSVA R package (v1.32.0). We then tested each pair of conditions for a difference in the GSVA enrichment scores of each hallmark pathway. We used a simple linear model and *t* statistics implemented in the limma[87] R package (v3.38.3) that uses an empirical Bayes shrinkage method. The Benjamini–Hochberg method[69] was used to adjust *p* values for multiple hypothesis testing within the false-discovery framework.

*Cell-type annotation.* The R package *SingleR* v1.0.1[88] was used to annotate cell types based on correlation profiles with bulk RNA-seq from the Immunological Genome Project (ImmGen) database[89]. Cell-type annotation was performed for individual cells of the whole dataset, as well as the lymphocyte and myeloid cell subsets of the dataset.

*Comparison to normal muscle macrophages.* To compare macrophages from transplant and primary tumors to macrophages from normal muscle, the Seurat label transfer method[82] was used to map data from previously published data sets of macrophages resident in normal muscle[38,39] onto myeloid cells from primary and transplant sarcomas. These two sets of normal muscle macrophages were extracted from published 10x scRNA-seq data sets containing cells from healthy murine limb muscle: (1) 308 macrophage cells from Tabula Muris et al.[38] (R object available on Figshare: https://doi.org/10.6084/m9.figshare.5821263.v1); and (2) 212 macrophage cells from Giordani et al.[39] (GEO accession code GSE110878). Cluster and sample labels from our dataset (reference) were transferred to macrophages from healthy limb muscle (query). *Seurat* projects the PCA structure of the reference onto the query. After finding anchors between the reference and query dataset, the *TransferData* function was used to classify the query cells based on a vector of reference cluster or sample labels. *TransferData* returns a matrix with predicted IDs and prediction scores, and the percentage of myeloid cells from the published data sets that fell into each myeloid cell reference cluster was calculated.

**Immunofluorescence staining.** Sarcomas were harvested after euthanasia, fixed in 4% PFA overnight, and preserved in 70% ethanol until paraffin embedding. Formalin-fixed paraffin-embedded tissues were stained with hematoxylin and eosin (H&E) on a Tissue-Tek Prisma autostainer (Sakura Finetek USA, Torrance, CA) to identify target tissue (tumor) locations. Target marks made on H&E slides were

then transferred to the surface of the corresponding tumor blocks. Two-millimeter cores were removed in triplicate from each tumor block using the semi-automated Pathology Devices TMArrayer™ instrument. The cores were transferred to blank paraffin recipient blocks to generate three recipient tissue microarray (TMA) blocks. The TMA grid layout consisted of 7 rows by 12 columns with 2.25 mm of core spacing as measured from the center of one core to the center of the next closest core. Four-micron sections were cut from TMAs, mounted onto positively-charged slides, and baked at 60 °C for 1 h. A Leica BOND Rx autostainer (Leica, Buffalo Grove, IL) was used to dewax and stain slides using Leica Bond reagents for dewaxing (Dewax Solution), retrieval of antigens, and stripping of antibodies (Epitope Retrieval Solution 2), and rinsing after each step (Bond Wash Solution). After the secondary and tertiary applications using high-salt TBST solution (0.05 M Tris, 0.3 M NaCl, and 0.1% Tween-20, pH 7.2–7.6), a high stringency wash was performed.

Antibody stripping and antigen retrieval steps were performed at 100 °C and all other steps were performed at ambient temperature. Three percent of $H_2O_2$ was applied for 8 min to block endogenous peroxidase, followed by protein blocking with TCT buffer (0.05 M Tris, 0.15 M NaCl, 0.25% Casein, 0.1% Tween-20, pH 7.6 +/− 0.1) for 30 min. The first primary antibody (Position 1, Supplementary Table 2) was applied for 60 min, followed by application of the secondary antibody for 10 min, then application of the tertiary TSA-amplification reagent (OPAL fluor, Akoya Biosciences, Menlo Park, CA) for 10 min. The primary and secondary antibodies were stripped with retrieval solution for 20 min. Then, beginning with application of 3% $H_2O_2$, the process was repeated with the second primary antibody (Position 2, Supplementary Table 2). The process was repeated until all positions were completed, but no stripping was performed after the final position. After removal from the autostainer, slides were stained with Spectral DAPI (Akoya) for 5 min, rinsed for 5 min, then coverslipped with Prolong Gold Antifade reagent (Invitrogen/Life Technologies, Grand Island, NY). Slides were cured for 24 h at room temperature in the dark, then all cores were acquired using the Akoya Vectra 3.0 or Akoya Polaris (MOTIF) Automated Imaging Systems for Panel 1 or Panel 2 slides (Supplementary Table 2), respectively. Akoya Phenoptics inForm software was used to process the images into multi-image TIFFs for use in HALO image analysis software (Indica Labs, Corrales, NM).

**Statistics and study design.** Experiments were designed such that littermate controls were used for all experiments. Statistical tests performed are indicated in figure legends. The experiments were randomized, and investigators were blinded to treatment during measurement and data collection. No statistical methods were used to predetermine sample size. Measurements were taken from distinct samples; the same sample was not measured repeatedly.

**Reagent information.** See Supplementary Tables 1 and 2 for reagent catalogue numbers, antibody clones, and dilutions.

**Reporting summary.** Further information on research design is available in the Nature Research Reporting Summary linked to this article.

## Data availability

All sequencing data generated for this manuscript have been deposited in publicly accessible databases. The primary and transplant tumor bulk RNA-seq data generated in this study are available in the NCBI Gene Expression Omnibus (GEO) database under the accession code GSE148856. The primary and transplant tumor and liver whole-exome sequencing data generated in this study are available in the Bioproject database under the accession code PRJNA556574. The scRNA-seq data generated in this study are available in the SRA database under the accession code PRJNA556477. The *Rag2*[−/−] and *Rag2*[+/−] bulk tumor RNA-seq data generated in this study are available in the NCBI GEO database under the accession code GSE154874. The *Rag2*[−/−] and *Rag2*[+/−] tumor and tail whole-exome sequencing generated in this study are available in the Bioproject database under the accession code PRJNA630870. The mass cytometry data generated in this study are available in the flowrepository.org database under the ID code FR-FCM-Z28C. The Tabula Muris Consortium macrophage data used in this study are available in the Figshare database [https://doi.org/10.6084/m9.figshare.5821263.v3]. The Giordani et al.[39] macrophage data used in this study are available in the NCBI GEO database under the accession code GSE110878. The remaining data are available within the Article, Supplementary Information or available from the authors upon request. Correspondence and requests for materials should be addressed to Y.M.M. or D.G. K. Source data are provided with this paper.

## Code availability

Computer codes used to generate survival curves, Fig. 4a, Supplementary Fig. 3a–c, and downstream scRNA-seq analysis in this manuscript can be found at https://gitlab.oit.duke.edu/wisdom2020/NatureCommunications. The analysis for Fig. 2 and Supplementary Fig. 1 was performed with a proprietary pipeline and we are unable to publicly release this code. However, all raw data including those used to generate Fig. 2 and Supplementary Fig. 1 have been made publicly available and the implementation details in the "Methods" and Supplementary Information allow for independent replication of these results.

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

## Acknowledgements

We thank Anton Berns (Netherlands Cancer Institute) for providing the *Trp53^{fl/fl}* mice and Tyler Jacks (MIT) for providing the 129/SvJae mice. Mass cytometry analysis was performed at the University of North Carolina Mass Cytometry Core Lab, which is supported by the University Cancer Research Fund (UCRF) and the UNC Cancer Center Core Support Grant (P30CA016086). We thank David Corcoran (Duke Genomic Analysis and Bioinformatics Shared Resource) for providing valuable bioinformatics support and feedback. We thank MSD for providing muDX400 and isotype control antibodies. We thank Charles Drake for providing critical feedback on this manuscript. This work was supported by awards F30CA221268 (A.J.W.), ASCO Young Investigator Award (Y.M.M.), U24 CA220245 (Y.M.M.), RSNA Research Resident/Fellow Grant (Y.M.M.), 1R38CA245204 (C.L.K.), Sarcoma Alliance for Research through Collaboration (SARC) SPORE 5U54CA168512 (D.G.K. and Y.M.M.), T32GM007171 (A.J.W. and J.E.H.), R35CA197616 (DGK), The Leon Levine Foundation (D.G.K.), the Emerson Collective (D.G.K.), the Duke Cancer Center Support Grant (P30CA14236), and P01CA142538 (K.O.) from the National Cancer Institute. The funders had no role in the design of the study, collection, analysis, and interpretation of the data or writing the manuscript.

## Author contributions

A.J.W., Y.M.M., and D.G.K. conceived the project, designed experiments, and wrote the manuscript. A.J.W., Y.M.M., C.S.H., J.E.H., E.S.X., D.J.C., C.L.K., and L.L. bred and genotyped mice. A.J.W., Y.M.M., C.S.H., J.E.H., R.P., A.M.B., D.J.C., C.L.K., and E.S.X. induced and measured sarcomas and treated mice with antibodies. A.J.W., Y.M.M., C.S.H., R.P., A.M.B., and E.S.X. performed surgery on mice. A.J.W., Y.M.M., and N.W. performed radiation treatments. A.J.W., C.S.H., R.P., and A.M.B. harvested tissue, generated cell lines, and extracted DNA and RNA for RNA-seq and WES. A.J.W. and J.E.H. performed bone marrow transplants. X.Q., D.Z., and K.O. performed analysis of mouse survival data, bulk RNA-seq pathway enrichment, and deposited the RNA-seq and WES data to publicly available databases. B.Y.N., D.A.K., A.A.A., and M.D. analyzed T.C.G.A. sarcoma data and performed CIBERSORTx analysis and human-mouse comparisons. L.C. and E.S.M. performed analysis of RNA-seq data and WES data for mutational load and neoantigen analysis. A.J.W. performed and analyzed CyTOF and collected and processed tumor cells for scRNA-seq. T.B. performed and analyzed scRNA-seq data and uploaded scRNA-seq data to the SRA database. H.F. performed GSVA analysis on scRNA-seq data and comparison of scRNA-seq on macrophages from normal muscle vs tumor. A.J.W. and C.S.H. collected tumors for histology and Y.M. processed tissue for histology. K.S.S. performed immunofluorescence staining and A.J.W. performed quantification. All authors reviewed the manuscript.

## Competing interests

D.G.K. is a cofounder of XRAD Therapeutics, which is developing radiosensitizers. D.G.K. and Y.M.M. are recipients of a Stand Up To Cancer (SU2C) MSD Catalyst Grant studying pembrolizumab and radiation therapy in sarcoma patients. D.G.K. has received research funding from XRAD Therapeutics, Eli Lilly & Co., Bristol Myers Squibb, Varian Medical Systems, and served as chair of the Developmental Therapeutics Committee for the Sarcoma Alliance for Research through Collaboration. E.S.M. and L.C. are employees of Merck Sharp & Dohme Corp, a subsidiary of Merck & Co., Inc., Kenilworth, NJ, USA. The other authors declare no competing interests.
