## [Peer Review File · Nature Communications]

REVIEWERS' COMMENTS

Reviewer #4 (Remarks to the Author):

Wisdom, et al. demonstrate convincingly that autochthonous models of soft-tissue sarcoma, developing from the point of induced somatic mutations, co-evolve along with the host immune system in such a way that these tumors are rather "cold" tumors, similar to the vast majority of human soft-tissue sarcomas. These same tumors, when transplanted into a syngeneic host of the same strain, but not to a second site on the same mouse, develop more of an inflamed immune response, fitting with the very small subset of human sarcomas that appear to be responsive to immune check-point inhibitors. All of the neoantigen profiling, immune infiltrates profiling, and especially the experiments with bone marrow transplant effort to reset the host immune system, were elegant experiments, driving home the very simple, but very important point (especially for all of us who engage in genetic model generation in mice) that tumors arising from somatic genetic initiation are very different from, and much more realistic than transplanted or xenografted tumors. There is more than just an intact immune system (which can obviously also be modeled with syngeneic transplants); there is a full biology of sarcomagenesis at work. I think this is superb work and moves the field forward for sarcoma clinical care, but more importantly for the pre-clinical modeling of sarcoma care. We can now choose models that are more appropriate to the scenarios we are trying to develop treatments to address.

Please find the comments from Reviewer #4 below in full (blue text) with our point-by-point response (black text).

Reviewer #4 (Remarks to the Author):

Wisdom, et al. demonstrate convincingly that autochthonous models of soft-tissue sarcoma, developing from the point of induced somatic mutations, co-evolve along with the host immune system in such a way that these tumors are rather "cold" tumors, similar to the vast majority of human soft-tissue sarcomas. These same tumors, when transplanted into a syngeneic host of the same strain, but not to a second site on the same mouse, develop more of an inflamed immune response, fitting with the very small subset of human sarcomas that appear to be responsive to immune check-point inhibitors. All of the neoantigen profiling, immune infiltrates profiling, and especially the experiments with bone marrow transplant effort to reset the host immune system, were elegant experiments, driving home the very simple, but very important point (especially for all of us who engage in genetic model generation in mice) that tumors arising from somatic genetic initiation are very different from, and much more realistic than transplanted or xenografted tumors. There is more than just an intact immune system (which can obviously also be modeled with syngeneic transplants); there is a full biology of sarcomagenesis at work.

I think this is superb work and moves the field forward for sarcoma clinical care, but more importantly for the pre-clinical modeling of sarcoma care. We can now choose models that are more appropriate to the scenarios we are trying to develop treatments to address.

We thank Reviewer #4 for the positive feedback on our manuscript and we agree that the revised manuscript "moves the field forward for... the pre-clinical modeling of sarcoma care."